# Dynamic Scenario Analysis of Science and Technology Innovation to Support Chinese Cities in Achieving the “Double Carbon” Goal: A Case Study of Xi’an City

**DOI:** 10.3390/ijerph192215039

**Published:** 2022-11-15

**Authors:** Renquan Huang, Jing Tian

**Affiliations:** 1School of Economics and Finance, Xi’an International Studies University, Xi’an 710128, China; 2FinTech Innovation Research Center, Xi’an International Studies University, Xi’an 710128, China

**Keywords:** science and technology innovation, carbon emissions, carbon peak, carbon neutrality, Monte Carlo simulation

## Abstract

Since the Chinese government proclaimed the “double carbon” goal in 2020, carbon emissions reduction has become an important task for the Chinese government. Cities generate more than 60% of carbon emissions. There are many challenges in achieving the “double carbon” goal for the cities of China. Science and technology innovation (STI) provides a feasible path, and the mechanism of STI influencing carbon emissions is analyzed. The STI factors, economic factors, energy factors, and population factors are studied based on the generalized Divisia index method. According to the decomposing results, science and technology innovation investment is the most important increasing factor in carbon emissions, and technology innovation investment efficiency is the most important decreasing factor, respectively. Three scenarios are set up and simulated with Monte Carlo technology evaluating the city of Xi’an in China. Under the baseline development scenario, it cannot achieve the carbon peak goal, and the uncertainty of carbon emissions increases. Under the green development scenario, it will peak in 2051, with a 95% confidence interval of 6668.47–7756.90 × 10^4^ tons. Under the technology breakthrough scenario, the lower and median boundaries of carbon emissions peak at 4703.94 × 10^4^ tons and 4852.39 × 10^4^ tons in 2026, and the upper boundary peaks at 5042.15 × 10^4^ tons in 2030. According to the Environmental Kuznets Curve theory, it will peak between 2028 and 2029 with a GDP per capita of CNY 153,223.85. However, it will fail to achieve the carbon neutrality goal by 2060, and should rely on the national carbon trading market of China to achieve the goal with a trading volume of 2524.61–3007.01 × 10^4^ tons.

## 1. Introduction

Greenhouse gas emissions have attracted extensive attention all over the world. In September 2020, China proclaimed that it would aim to reach peak carbon emissions by 2030 and would work towards achieving carbon neutrality by 2060 (the “double carbon” goal). Therefore, the CPC Central Committee and the State Council issued “Opinions on Complete and Accurate Implementation of the New Development Concept to Do a Good Job in Carbon Peak and Carbon Neutral Work” [1] and approved “Action Plan for Carbon Dioxide Peaking Before 2030” [2].

The relationship between urban development and climate warming is becoming stronger. Cities are both a major contributor to climate warming and a major locus of carbon emissions reduction, generating more than 60% of carbon emissions [3]. Urban carbon emission reduction is the key to low-carbon development in China and even the world. Urban carbon peak is the premise to achieve the “double carbon” goal [4]. Therefore, the Chinese government has undertaken a series of efforts to enable cities to achieve the “double carbon” goal. However, because China is still in a period of industrialization and urbanization, the pattern of energy consumption, mainly coal, has not yet undergone a fundamental change [5]. The total energy consumption will continue to climb in the future. Thus, it is difficult to immediately reduce carbon emissions in cities, and there is a contradiction between urban economic development and carbon emission reduction. Although some cities have achieved success in carbon emission reduction, most cities are still facing tremendous pressure to reduce emissions. There are many challenges in achieving the “double carbon” goal for the cities of China.

As an important tool for energy conservation and emission reduction, science and technology innovation (STI) has received extra attention from the Chinese government. In August 2022, nine ministries in China, including the Ministry of Science and Technology, issued the “Science and Technology to Support the Implementation Plan of Carbon Peak and Carbon Neutrality (2022–2030)” [6]. It points out that the supporting role of STI is extremely important to guarantee high-quality economic development and the achievement of the “double carbon” goal. On the one hand, STI could promote the development of new energy technologies. While new energy and traditional energy are mutual alternatives, the use of new energy can reduce the demand for traditional fossil fuel energy [7]. On the other hand, STI could improve the efficiency of traditional energy technologies. It would optimize the traditional production process, and reduce the process’ carbon emissions level [8].

STI provides a feasible path for cities to achieve the “double carbon” goal. However, there are still some shortcomings in the existing research on the role of STI in supporting Chinese cities to achieve the “double carbon” goal. These are as follows: First, the roles of urban carbon emission [9,10] and STI on carbon emission reduction [11,12] were studied separately. There is a lack of comprehensive research on STI to support cities in achieving the “double carbon” goal. Second, most of the existing research failed to include the STI factor in the generalized Divisia index method (GDIM) [13,14], and STI was not as important a factor as economics, energy, and population in influencing carbon emissions. Third, research on the projection of urban carbon emissions is not enough, especially considering the length of the period. Some scholars have studied carbon emissions and the time to 2030 [10,15]; however, the projection of carbon neutrality by 2060 was not enough.

Therefore, we have studied the path of STI to support cities achieving the “double carbon” goal and take the city of Xi’an in China as a study case. This paper tries to better understand the impact of STI on carbon emissions. First, this paper analyzes the dual effect of STI on carbon emissions and explores multiple paths for STI to influence carbon emissions. Second, based on GDIM, this paper discusses the influential factors on carbon emission, including STI factors (STII, STIICI, and STIIE), economic factors (GDP and OCI), energy factors (EC, ECCI, and EI), and population factors (population and CPC). Third, the baseline development scenario, the green development scenario, and the technology breakthrough scenario are set, respectively. Monte Carlo technology is applied to simulate the carbon emissions in Xi’an according to the scenarios.

The remainder of this paper is structured as follows: Section 2 reviews the existing literature. Section 3 introduces the research methods. Section 4 analyzes the results of GDIM decomposition and scenario simulation, and discusses the technology breakthrough scenario with Environmental Kuznets Curve (EKC) theory. Section 5 summarizes the paper and provides policy suggestions.

## 2. Literature Review

### 2.1. Evaluation and Decomposition Methods of Carbon Emissions

According to the existing research on the evaluation of urban carbon emissions, the method based on the Intergovernmental Panel on Climate Change (IPCC) framework was widely accepted [16]. In 2021, China’s Ministry of Ecology and Environment (2021) [17] issued “Guidelines for the Preparation of Provincial Carbon Emission Peaking Action Plans”. It has made some additions to the IPCC guidelines according to the situation in China. There is some research on the evaluation of carbon emissions in Xi’an [18,19,20]. In this paper, five sectors were selected to evaluate the carbon emissions of Xi’an, comprising energy, industrial processes and product use, agriculture, waste, and forestry and other land use.

As environmental problems become more and more prominent, structural decomposition analysis (SDA) [5,21] and index decomposition analysis (IDA) are widely used to study the driving factors in carbon emissions. The SDA is comparatively more data-intensive and needs to be built on complex input–output tables. The input–output tables are updated every five years in China, and the IDA is more convenient for this research. IDA methods mainly include the Laspeyres Index (LI) decomposition method and the Divisia Index decomposition method. Chen et al. (2022) [22] used the LI decomposition method to discuss the effects of economic growth and coal intensity on China’s coal consumption. There were 0 value or residual problems with the LI [23]. The Logarithmic Mean Divisia Index (LMDI) could overcome the above-mentioned defects, and many recent studies have employed LMDI to analyze factors that impact CO_2_ emissions [24,25,26]. The IDA methods (including LI, LMDI, etc.) rely on the Kaya identity and express the explanatory variables as the product of several factors while ignoring the dependence between the multiplied factors [27]. It may lead to contradictory decomposition results if different factors are selected for the decomposition model [13]. Given the defects above, the GDIM was proposed [28] and could overcome the defects of the existing index decomposition methods mentioned above. Since the GDIM was proposed, scholars have conducted empirical research on its effectiveness [14,15].

### 2.2. Impact Mechanism of Technological Innovation on Carbon Emissions

Based on the existing literature, Table 1 presents a summary of the decomposition methods and driving factors of carbon emissions. Accordingly, scholars have usually taken economic activity (GDP), energy intensity, energy consumption, energy structure, population, etc., as the key driving factors affecting carbon emissions. Others have found that the energy and carbon emissions are significantly affected by STI [13]. STI has a dual effect on carbon emissions. The effect of technical progress on carbon emission intensity was negative in eastern and western China, but it was positive in central China [29]. On the one hand, STI can have a negative effect on carbon emissions. Nguyen et al. (2020) [11] studied the G20 countries, and the results showed there was a negative relationship between STI and carbon emissions. Sun et al. (2020) [12] based their study on a moderated mediation model and found that industrial structure upgrading and technology innovation could significantly reduce carbon emissions. Furthermore, green technology innovation is the most important part of STI in carbon emissions reduction [30]. Green technology innovation and institutional quality were identified as effective mechanisms to mitigate carbon emissions and promote sustainable development [8]. On the other hand, STI can have a positive effect on carbon emissions. Liu et al. (2021) [31] studied the threshold effects of energy consumption, technology innovation, and supply chain management on enterprise performance in China’s manufacturing industry, and found that technology innovation had a dual effect on enterprise performance.

Figure 1 shows the mechanism of STI influencing carbon emissions, and the acronyms are in Table 2. Economic expansion is the primary contributor to carbon emissions in China [32], and STI is an important driver of economic expansion [33]. Government financial expenditure on science and technology is the most common and focused part of science and technology innovation investment (STII). Science and technology innovation investment efficiency (STIIE) measures the economic growth per unit of STII funds, and affects carbon emissions through its effect on economic growth. Huang et al. (2021) [34] studied the science and technology investment efficiency of Xi’an during 2000–2019 and found it had been greatly improved. Science and technology innovation investment carbon intensity (STIICI) measures the CO_2_ produced per unit of STII, and could directly reflect the carbon emission levels of the investment.

## 3. Methods

### 3.1. Decomposition of Carbon Emission Factors Based on GDIM

The GDIM uses the Kaya identity to construct a multifactor decomposition model and reveals the driving factors of carbon emissions. According to the GDIM, carbon emissions can be decomposed into:(1)$$CO_{2}=\left(CO_{2}/GDP\right)\times GDP=\left(CO_{2}/E\right)\times E=\left(CO_{2}/P\right)\times P=\left(CO_{2}/T\right)\times T$$
(2)$$GDP/P=\left(CO_{2}/P\right)\times \left(CO_{2}/GDP\right)$$
(3)$$E/GDP=\left(CO_{2}/GDP\right)\times \left(CO_{2}/E\right)$$

In this paper, we need to symmetrically incorporate all of them into the factor analysis. Considering the readability, we make the following denomination: $Z=CO_{2}$, $X_{1}=GDP$, $X_{2}=CO_{2}/GDP$, $X_{3}=E$, $X_{4}=CO_{2}/E$, $X_{5}=T$, $X_{6}=CO_{2}/T$, $X_{7}=P$, $X_{8}=CO_{2}/P$, $X_{9}=GDP/T$, $X_{10}=E/GDP$. The meaning of the variables are shown in Table 2. Therefore, Equations (1)–(3) could be rewritten as:(4)$$Z=X_{1}X_{2}=X_{3}X_{4}=X_{5}X_{6}=X_{7}X_{8}$$
(5)$$X_{9}=X_{6}/X_{2}$$
(6)$$X_{10}=X_{2}/X_{4}$$

To apply the GDIM, we make a further transformation based on Equations (4)–(6):(7)$$Z=X_{1}X_{2}$$
(8)$$X_{1}X_{2}-X_{3}X_{4}=0$$
(9)$$X_{1}X_{2}-X_{5}X_{6}=0$$
(10)$$X_{1}X_{2}-X_{7}X_{8}=0$$
(11)$$X_{1}-X_{5}X_{9}=0$$
(12)$$X_{3}-X_{1}X_{10}=0$$

According to GDIM, the gradient of $Z\left(X\right)$ and the Jacobian matrix are obtained as follows:(13)$$\nabla Z=\langle X_{2},X_{1},0,0,0,0,0,0,0,0\rangle^{T}$$
(14)$$\Phi_{X}=\left[\begin{array}{cccccccccc} X_{2} & X_{1} & -X_{4} & -X_{3} & 0 & 0 & 0 & 0 & 0 & 0\\ X_{2} & X_{1} & 0 & 0 & -X_{6} & -X_{5} & 0 & 0 & 0 & 0\\ X_{2} & X_{1} & 0 & 0 & 0 & 0 & -X_{8} & -X_{7} & 0 & 0\\ 1 & 0 & 0 & 0 & -X_{9} & 0 & 0 & 0 & -X_{5} & 0\\ -X_{10} & 0 & 1 & 0 & 0 & 0 & 0 & 0 & 0 & -X_{1} \end{array}\right]$$

Then, we reach the following formula:(15)$$∆Z\left[X\left|\Phi \right.\right]=\int_{L}\nabla Z^{T}\left(I-\Phi_{X}\Phi_{X}^{+}\right)dX$$
where L is the time range from $t_{0}$ to $t_{1}$. I is the identity matrix. $\Phi_{X}^{+}$ is the generalized matrix of $\Phi_{X}$. If the columns of $\Phi_{X}$ are linearly independent, then $\Phi_{X}^{+}={\left(\Phi_{X}^{T}\Phi_{X}\right)}^{-1}\Phi_{X}^{T}$.

The quantitative factors $X_{1}=GDP$,$X_{3}=E$,$X_{5}=T$ and $X_{7}=P$ are considered given functions of a model time t. According to Vaninsky (2014) [28], the range of time change does not affect the final result. Therefore, we assume 0≤t≤1 and an exponential change of $X_{1}$, $X_{3}$, $X_{5}$ and $X_{7}$.
(16)$$Q\left(t\right)={\left(\frac{Q_{1}}{Q_{0}}\right)}^{t}$$

Q stands for any quantitative or relative indicator $X_{i}$ or Z. $Q_{0}$ and $Q_{1}$ are the base and final values. We reach:(17)$$\frac{dQ\left(t\right)}{dt}=Q\left(t\right)\cdot ln\left(\frac{Q_{1}}{Q_{0}}\right)$$

### 3.2. The Model of Monte Carlo Simulation

According to the decomposition results based on GDIM in Section 4.1, STII (T), EO (GDP), and EC (E) are important factors contributing to the growth of carbon emissions, while STIIE (GDP/T) is the most important decreasing factor. Therefore, Xi’an should pay more attention to the factors mentioned above to achieve the “double carbon” goal. Carbon emissions could be rewritten according to the key factors:(18)$$CO_{2}=T\times \frac{GDP}{T}\times \frac{E}{GDP}\times \frac{CO_{2}}{E}$$

We defined the rate of STII (T), STIIE (GDP/T), EI (E/GDP), ECCI ($CO_{2}/E$), and CE ($CO_{2}$) as α, β, δ, ϕ, and ω. Consequently, there are Formulas (19)–(23):(19)$$CO_{2,t+1}=CO_{2,t}\left(1+\omega \right)$$
(20)$$T_{t+1}=T_{t}\left(1+\alpha \right)$$
(21)$${\left(GDP/T\right)}_{t+1}={\left(GDP/T\right)}_{t}\left(1+\beta \right)$$
(22)$${\left(E/GDP\right)}_{t+1}={\left(E/GDP\right)}_{t}\left(1+\delta \right)$$
(23)$${\left(CO_{2}/E\right)}_{t+1}={\left(CO_{2}/E\right)}_{t}\left(1+\phi \right)$$

Substituting Equations (19)–(23) into Equation (18) yields Equation (24):(24)$$CO_{2,t}\left(1+\omega \right)=T_{t}\left(1+\alpha \right)\cdot {\left(GDP/T\right)}_{t}\left(1+\beta \right)\cdot {\left(E/GDP\right)}_{t}\left(1+\delta \right)\cdot {\left(CO_{2}/E\right)}_{t}\left(1+\phi \right)$$

The rate of carbon emissions ω could be expressed as follows:(25)$$\omega =\left(1+\alpha \right)\cdot \left(1+\beta \right)\cdot \left(1+\delta \right)\cdot \left(1+\phi \right)-1$$

According to Equation (25), we can apply the Monte Carlo simulation to predict carbon emissions in the future of Xi’an. Before Monte Carlo simulation, the probability distribution of variables needs to be determined, and triangular distribution [37] is chosen in this paper. Sampling is undertaken according to the median, minimum, and maximum values of the different variables according to the scenarios in Section 3.3. The sample number is set to 100,000 times for each variable.

### 3.3. Scenario Setting

To simulate the “double carbon” path of Xi’an, three scenarios are set, including the baseline development scenario, the green development scenario, and the technology breakthrough scenario. These scenarios are based on the existing emission reduction policies and the potential emission reduction of Xi’an.

#### 3.3.1. Baseline Development Scenario

Carbon emissions in 2021 have already occurred, but most of the data are unavailable due to the statistical yearbooks not yet being published. Therefore, we cannot evaluate the accurate carbon emissions according to the model. The “National Economic and Social Development Statistical Bulletin of Xi’an in 2021” [38] report shows that the regional GDP in 2021 was CNY 10,688.28 × 10^8^. The energy consumption of large industrial enterprises was 584.77 × 10^4^ tons of standard coal, a decrease of 2.30%. The electricity consumption of society was 489.37 × 10^8^ Kw⋅h, an increase of 17.90%. In this regard, it could be estimated that the energy consumption of Xi’an in 2021 was 2854.86 × 10^4^ tons of standard coal, and the EI (E/GDP) rate was −5.11%. Since the data for carbon emissions and STII were not available, the trend extrapolation method was adopted to estimate 4643.81 × 10^4^ tons and CNY 290,297.50 × 10^4^, respectively. In this regard, the estimated rate of STII (T), STIIE (GDP/T), and ECCI ($CO_{2}/E$) were 24.76%, −14.50%, and −6.60% in 2021.

In the baseline development scenario, the economic and societal policies of Xi’an are based on past development characteristics, and the technology innovation environment is assumed to remain unchanged. The rate of factors, including the STII (T), STIIE (GDP/T), ECCI ($CO_{2}/E$), etc., maintain the original trend. During the period 2022–2060, the rate of each factor is set based on their average annual rate in the following five periods: 1995–2020, 2000–2020, 2005–2020, 2010–2020, and 2015–2020. The maximum and minimum rates during five periods are taken as the maximum and minimum values accordingly. The median values are determined by Equation (26):(26)$${\bar{V}}_{i}=\sum_{k}r_{i,k}V_{i,k}$$
where ${\bar{V}}_{i}$ is the median value of the ith variable. $r_{i,k}$ is the weight of the ith variable in the kth period: 0.4 (2015–2020), 0.3 (2010–2020), 0.15 (2005–2020), 0.1 (2000–2020), and 0.05 (1995–2020). $V_{i,k}$ is the rate of the ith variable in the kth period. According to (26), the parameters are shown in Table 3 for the baseline development scenario.

#### 3.3.2. Green Development Scenario

In the green development scenario, it is assumed that the government has strengthened the measures against climate change, which will lead to the optimization of energy structure, the improvement of energy-saving technology, the enhancement of capital productivity, and the steady growth of investment in STI. Significant results have been achieved in resource conservation, clean production and consumption, and circular economy. Green development has become the consensus of the whole society.

For science and technology innovation investment (T), there is no clear indicator. According to “The Outline of the 14th Five-Year Plan for Economic and Social Development and Long-rang Objectives Through the Year 2035 of Shaanxi Province” A [39], it specified the average annual rate of R&D expenditure during the “14th Five-Year Plan (2021–2025)” will be more than 8.00%. It is set as the median value in the period. For the period 2026–2060, a slight decrease in the rate is assumed, and the median values are set as shown in Table 4. Considering the effectiveness and uncertainty of policy implementation, the maximum and minimum rates are increased or decreased by 1.00% from the median value, respectively.

Regarding science and technology innovation investment efficiency (GDP/T), according to “The Outline of the 14th Five-Year Plan for Economic and Social Development and Long-rang Objectives Through the Year 2035 of Xi’an” [40], the GDP growth average annual rate during the “14th Five-Year Plan” will be no less than 6.50%, while in 2025, GDP will reach CNY 14,000 × $10^{8}$ with an average annual rate of 6.92% compared to 2020. Combined with the average annual rate of STII (T) at 8.00%, the average annual rate of STIIE (GDP/T) is calculated to be −1.00% during the “14th Five-Year Plan” period. It is assumed that the GDP average annual rate is 6.50% during the “15th Five-Year Plan (2026–3030)” period, and the average annual rate of STIIE (GDP/T) is calculated to be −0.47%. If the GDP average annual rate decreases by 1.00% every 10 years during the period of 2031–2040, 2041–2050, and 2051–2060, the average annual rate values of STIIE (GDP/T) are −0.47%, −0.48%, and −0.48%, respectively. The maximum and minimum average annual rates are increased or decreased by 1.00% from the median value.

Regarding energy intensity (E/GDP), according to “The Outline of the 14th Five-Year Plan (2021–2025) for Economic and Social Development and Long-rang Objectives Through the Year 2035 of Xi’an” [40], energy intensity will take the national or provincial indicator. The “Action Plan for Carbon Dioxide Peaking Before 2030” [2] specifies that the energy consumption per unit of GDP until 2025 should decrease by 13.50% compared with 2020. In this regard, it can be estimated that the average annual rate of EI (E/GDP) is −2.86% during the “14th Five-Year Plan” period, which is set as the median value. The average annual rate of EI (E/GDP) is assumed to remain at −2.86% during 2026–2060, when Xi’an will continue its energy conservation policies. The maximum and minimum average annual rates are increased or decreased by 1.00% from the median value, respectively.

Regarding energy consumption carbon intensity ($CO_{2}/E$), the “Action Plan for Carbon Dioxide Peaking Before 2030” [2] clarified that: by 2025, the share of non-fossil fuels in total energy consumption will reach around 20.00%, while carbon emissions per unit of GDP will drop by 18.00% compared with 2020 levels; and by 2030, the share of non-fossil energy consumption will reach around 25.00%, and carbon emissions per unit of GDP will have dropped by more than 65.00% compared with 2005, successfully achieving carbon peak before 2030. From 2022 to 2025, the proportion of non-fossil energy consumption will increase by about 1.00% annually, and the average annual rate of ECCI is calculated to be −1.06%. This is assuming that from 2026 to 2060 Xi’an continues the energy policies, and the median rates are shown in Table 4 for different periods. The maximum and minimum average annual rates are increased or decreased by 0.2% from the median value, respectively.

#### 3.3.3. Technology Breakthrough Scenario

The technology breakthrough scenario is based on the green development scenario, and the expected changes in STII (T), STIIE (GDP/T), EI (E/GDP), and ECCI ($CO_{2}/E$), etc., are enhanced to obtain low-carbon development.

Regarding science and technology innovation investment (T), under the technology breakthrough scenario, it maintains the same rate as in the green development scenario. More investment is used for the development of energy conservation and emission reduction technology, as well as the updating and upgrading of related equipment, while less investment is used to improve production efficiency and scale expansion.

Regarding science and technology innovation investment efficiency (GDP/T), the technology breakthrough scenario has less investment to improve productivity, which will lead to a subsequent decrease in output levels. Therefore, the economic output of the technological breakthrough scenario is lower compared to the green development scenario from a macro perspective. Considering the impact of the low-carbon development model on the economic growth rate [41], it is assumed that the average annual rate of STIIE (GDP/T) is 1.00% lower compared to the green development scenario.

Regarding energy intensity (E/GDP), the “Action Plan for Carbon Dioxide Peaking Before 2030” [2] clarified that the energy consumption per unit of GDP will drop 13.50% by 2025 compared with 2020. Xi’an is at the forefront of energy conservation and emission reduction in Shaanxi Province and even in China. As economic development and income levels increase, it becomes progressively more difficult to reduce the EI (E/GDP). Under the technology breakthrough scenario, the EI (E/GDP) target is set to decrease by 17.00% compared to 2020, obtaining an average annual rate of −3.66% during the “14th Five-Year Plan” period. In addition, it is assumed that the energy conservation and emission reduction policies will continue for the rest of the period, i.e., the average annual rate of EI (E/GDP) will remain −3.66%.

Regarding energy consumption carbon intensity ($CO_{2}/E$), due to the time lag between technology and the actual effect of energy conservation and emission reduction, it is assumed that the ECCI ($CO_{2}/E$) is consistent with the green development scenario during the “14th Five-Year Plan” period. With the development of new energy technology, it is assumed that the proportion of non-fossil energy consumption will increase from 25.00% to 28.00% by 2030, and the proportion will increase by 15.00% every 10 years during 2031–2060. Then the average annual rate of ECCI ($CO_{2}/E$) is calculated as shown in Table 5.

## 4. Results and Discussion

### 4.1. Decomposition Results of the GDIM

The data for Xi’an required for the GDIM decomposition between 1995 and 2020 are shown in Appendix A Table A1. Based on the GDIM, the results are analyzed by the cumulative rate for each factor. During this period, the cumulative value of carbon emission was 199.25%, shown in Figure 2a, with an average annual rate of 7.97%. After 2002, the rate of carbon emissions accelerated significantly in Xi’an. The cumulative value in 2002 was 42.32%, with an average annual rate of 6.05%; the cumulative growth from 2003 to 2020 was 156.93%, with an average annual rate of 8.81%.

Regarding the cumulative effect of economic factors, the economic factors include EO (GDP) and OCI ($CO_{2}/GDP$). The cumulative effects on carbon emissions during 1995–2020 are shown in Figure 2a. The cumulative contribution of EO (GDP) was 93.44%, with an average annual growth rate of 3.74%. It is the second most important increasing factor of carbon emissions after STII (T). The cumulative contribution of OCI ($CO_{2}/GDP$) was −35.84%, with an average annual decrease rate of 1.43%. It has a strong inhibitory effect on carbon emissions and becomes the second most important decreasing factor after STIIE (GDP/T).

Regarding the cumulative effect of energy factors, the energy factors include EC (E), ECCI ($CO_{2}/E$), and EI (E/GDP). The cumulative effects on carbon emissions during 1995–2020 are shown in Figure 2b. The cumulative contribution of EC (E) is 50.63%, with an average annual growth rate of 2.03%. It is an important contributor to carbon emissions. The cumulative contribution of ECCI ($CO_{2}/E$) is 11.32%, with an average annual growth rate of 0.45%. It reached a peak of 12.39% in 2010 and had been fluctuating downward since then. The cumulative contribution of EI (E/GDP) is −9.97%, with an average annual decrease rate of 0.39%. It is an important decreasing factor in carbon emissions.

Regarding the cumulative effect of population factors, the population factors include population (P) and CPC ($CO_{2}/P$). The cumulative effects on carbon emissions during 1995–2020 are shown in Figure 2c. The cumulative contribution of the population (P) is 10.78%, with an average annual growth rate of 0.43%, and shows a significant incremental trend. It is the lowest among all the increasing factors, indicating that the population scale contributes to carbon emissions, but the contribution is limited compared with other factors. The cumulative contribution of CPC ($CO_{2}/P$) is 40.76%, with an average annual growth rate of 1.63%. Before 2013, the cumulative contribution is 40.61%, and it has grown relatively fast. During 2013–2020, it shows a fluctuating growth trend, with a relatively limited growth rate of 0.15%.

Regarding the cumulative effect of science and technology innovation factors, the science and technology innovation factors include STII (T), STIICI ($CO_{2}/T$), and STIIE (GDP/T). The cumulative effects on carbon emissions during 1995–2020 are shown in Figure 2d. The cumulative contribution of STII (T) is 127.14%, with an average annual growth rate of 5.09%. It is the most important driving factor in carbon emissions. The cumulative contribution of STIICI ($CO_{2}/T$) is −1.64%, with an average annual decrease rate of 0.07%. The rates of STII (T) and STIICI ($CO_{2}/T$) have a strong negative correlation effect when comparing the curves. The cumulative contribution of STIIE (GDP/T) is −87.57%, with an average annual decrease rate of 3.50%, and shows a significant decline trend. It is the most important factor in carbon emissions reduction. The study shows that the economic development of Xi’an should be based on STI, accelerating the transformation and upgrading of the industrial structure to further improve quality and efficiency, and increasing technical support in the field of energy conservation and emission reduction.

### 4.2. Forecast Results of Scenario Simulation

By using MATLAB R2021a, Monte Carlo simulations were conducted for the baseline development scenario, the green development scenario, and the technology breakthrough scenario, and then the path of achieving the “double carbon” goal was scientifically selected based on the simulation results.

#### 4.2.1. Baseline Development Scenario

Under the baseline development scenario, the probability density plot of carbon emission evolution is shown in Figure 3a from 2022 to 2030. Figure 3b shows the upper, median, and lower boundaries of the 95% confidence interval during 2022–2060. Table 6 shows the average annual rates in different stages of three scenarios. In terms of carbon emissions, the simulated 95% confidence interval is 4363.56–5653.43 × 10^4^ tons in 2022, 5824.52–12,785.81 × 10^4^ tons in 2030, and increases up to 30,090.25–155,903.42 × 10^4^ tons in 2060. The confidence interval tends to diverge, indicating that the uncertainty of carbon emissions is increasing under the scenario. By analyzing the annual average rate of carbon emissions, the upper, median, and lower boundaries have maintained a faster rate of growth, with the upper boundary growing the fastest, followed by the median boundary and the lower boundary. The growth rate of the upper boundary tends to decrease at different stages, the median growth rate remains about 7.13%, while the growth rate of the lower boundary tends to increase. Under this scenario, if the past carbon emission reduction measures are maintained in the economic development pattern, Xi’an’s carbon emissions will further increase significantly. It means that Xi’an’s carbon emissions will not peak before 2030. The upper, middle and lower boundaries of carbon emissions will keep growing faster from 2022 to 2060. Under the baseline development scenario, Xi’an cannot achieve the “double carbon” goal. Therefore, Xi’an must adjust the energy conservation and emission reduction policies, and change the past development pattern, which puts high economic growth at the expense of the environment.

#### 4.2.2. Green Development Scenario

Under the green development scenario, the probability density plot of carbon emission evolution is shown in Figure 4a from 2022 to 2030. Figure 4b shows the upper, median, and lower boundaries of the 95% confidence interval during 2022–2060. In terms of carbon emissions, the simulated 95% confidence interval is 4706.59–4837.86 × 10^4^ tons in 2022, 5610.77–6095.95 × 10^4^ tons in 2030, and increases up to 6293.65–7486.55 × 10^4^ tons in 2060. Under this scenario, the upper, median, and lower boundaries peak in 2051 at the same time, with 6668.47, 7194.38, and 7756.90 × 10^4^ tons, respectively. The confidence interval tends to diverge. It indicates that the uncertainty of carbon emissions is increasing under the scenario, as well. By analyzing the annual average rate of carbon emissions, the growth rate of the upper, median, and lower boundaries gradually slows down, and the year 2051 is an inflection point for carbon emissions to reduce. During the period 2022–2051, the upper boundary grows the fastest, followed by the median boundary and the lower boundary. During the period 2051–2061, the lower boundary of carbon emissions decreases the fastest, followed by the median boundary and the upper boundary. Under the green development scenario, although Xi’an has taken certain measures to save energy and reduce emissions, STII in improving production efficiency and expanding production scale accounts for a relatively large proportion. It means that the carbon emissions fail to peak by 2030 and delay 21 years to 2051. It can be concluded that the rapid growth of carbon emissions can be effectively restrained if the government takes active energy conservation and emission reduction measures. However, there is still a big gap to achieve the “double carbon” goal.

#### 4.2.3. Technology Breakthrough Scenario

Under the technology breakthrough scenario, the probability density plot of carbon emission evolution is shown in Figure 5a from 2022 to 2030. Figure 5b shows the upper, median, and lower boundaries of the 95% confidence interval during 2022–2060. In terms of carbon emissions, the simulated 95% confidence interval is 4620.39–4749.77 × 10^4^ tons in 2022, 4632.58–5042.15 × 10^4^ tons in 2030, and 2524.61–3007.01 × 10^4^ tons in 2060. Under this scenario, both the lower and median bounds peak in 2026 at 4703.94 and 4852.39 × 10^4^ tons. The upper bound peaks in 2030 at 5042.15 × 10^4^ tons, respectively. The confidence interval shows a diverging trend in the period 2022–2045 and converges during 2045–2060. It indicates that the uncertainty of carbon emissions is manageable under this scenario. By analyzing the annual average rate of carbon emissions, the upper, median, and lower boundaries gradually decrease at a faster rate after reaching the peak, with the fastest decrease in the lower boundary, followed by the median boundary and the upper boundary. Under the technology breakthrough scenario, the carbon emissions of Xi’an will peak by 2030 and still fall short of the carbon neutrality goal by 2060. Combined with the situation of carbon sinks in Xi’an, it is impossible to achieve carbon neutrality by 2060 depending on its conditions. Therefore, it must rely on the national carbon trading market of China to achieve the “double carbon” goal, and the trading volume is about 2524.61–3007.01 × 10^4^ tons.

### 4.3. GDIM Decomposition of Technology Breakthrough Scenario

The results of the three scenarios conclude that the baseline development scenario cannot achieve the “double carbon” goal, the green development scenario peaks in 2051, and only the technology breakthrough scenario can peak by 2030. The goal of carbon neutrality would be achieved through carbon trading in the national carbon market by 2060. Therefore, the development path of technological breakthrough should be chosen for Xi’an, which increases investment in non-fossil energy use, energy conservation, and emission reduction technology.

To use the GDIM under the technological breakthrough scenario, the values of carbon emissions, regional GDP, energy consumption, and other factors can be obtained according to the Monte Carlo simulation during 2021–2060. The population rate of Xi’an is set according to Kang (2020) [42], where the average annual rate is 2.00% during 2022–2025, 1.00% during 2025–2030, −0.20% during 2030–2040, −0.35% during 2040–2050, and −0.45% during 2050–2060. It will begin to decline after 2030. During 2021–2060, the simulation results of each factor by median forecast are shown in Appendix A Table A1, and the results of decomposition by GDIM are shown in Appendix A Table A2.

From 2021 to 2025, carbon emissions will grow by 3.58% in Xi’an. EO (GDP) contributes 7.10%, and it is the most important contributing factor, followed by the STII (T) contributing 4.06%. EC (E) and population (P) contribute 3.38% and 2.00%, respectively. This indicates that the scale effect is still the main factor to promote carbon emissions, and the rate of carbon emissions is effectively controlled under the technology breakthrough scenario, which provides feasible conditions for carbon peak. OCI ($CO_{2}/GDP$) becomes the largest contributor of carbon emissions by −5.63%, followed by STIICI ($CO_{2}/T$) contributing −3.33%. In addition, the carbon intensity of EC ($CO_{2}/E$, −2.63%), CPC ($CO_{2}/P$, −1.09%), EI (E/GDP, −0.17%), and STIIE (GDP/T, −0.11%) also have a dampening effect on carbon emissions.

From 2026 to 2030, STII (T) is the most important increasing factor, contributing 6.70%, followed by the EO (GDP) contributing 4.06%. In addition, the STIICI ($CO_{2}/T$, −6.33%) surpassed the OCI ($CO_{2}/GDP$, −6.17%), becoming the most important decreasing factor.

From 2031 to 2040, the population (P, −0.47%) becomes a decreasing factor due to the negative population growth rate. During 2041–2050, EO (GDP, 21.48%) and OCI ($CO_{2}/GDP$, −17.19%) are the most important increasing and decreasing factors, respectively, and the contribution of EC (E, 1.42%) to carbon emissions is further reduced. From 2051 to 2060, carbon emissions are reduced by 23.12%. EC (E, −0.55%) becomes a decreasing factor, and the increasing factors are the EO (GDP, 6.90%) and the STII (T, 7.50%).

During 2021–2060, total carbon emissions are reduced by 40.66%. EO (GDP, 23.20%) and STII (T, 20.31%) are the two main increasing factors. ECCI ($CO_{2}/E$, −25.46%), OCI ($CO_{2}/GDP$, −24.33%) and STIICI ($CO_{2}/T$, −24.47%) are the three most important decreasing factors, respectively.

### 4.4. Environmental Kuznets Curve Effect

Named after Kuznets (1995) [43], the EKC hypothesis suggests that environmental degradation increases initially (environmental quality falls), and later falls (environmental quality increases) beyond a certain point with increasing per capita income along a country’s growth path. Emissions are a function of income, and there is an inverse U-shaped curve between emissions and income [44]. Based on the EKC research [45,46,47], the relationship between carbon emissions and GDP per capita (logarithm) is shown in Figure 6 during 1995–2060.

Equation (27) is obtained by the quadratic fitting function.
(27)$$y=-0.174x^{2}+4.155x-16.380$$
where y refers to carbon emissions *lnCO*_2_, x refers to income ln(GDP/P). $R^{2}$ is 0.9637 for Equation (27), indicating a good fit. According to the derivative of Equation (27), the extreme point of the curve can be calculated x=11.94, i.e., the GDP per capita is CNY 153,223.85. The GDP per capita is calculated to be CNY 106,968.37 in 2021, which is still on the left side of the inflection point according to the EKC hypothesis. Considering the trend of GDP per capita in Xi’an, the value is obtained between 2028 (CNY 152,014.96) and 2029 (CNY 160,293.01). Therefore, according to the EKC theory, carbon emissions will peak between 2028 and 2029 in Xi’an under the technology breakthrough scenario.

## 5. Conclusions

This paper focuses on the issue of STI to support Chinese cities in achieving the “double carbon” goal, and decomposes carbon emissions using the GDIM based on the mechanism of STI influencing carbon emissions. Exploring the path of STI to support carbon emission reduction, the baseline development, green development, and technology breakthrough scenarios are set up and simulated by a Monte Carlo simulation. The results are as follows:

Firstly, STI has an important impact on carbon emissions. STI affects carbon emissions through multiple pathways. During 1995–2020, the cumulative effect of carbon emissions is 199.25% based on the GDIM decomposition results. STII (T, 127.14%), EO (GDP, 93.44%), EC (E, 50.63%), CPC ($CO_{2}/P$, 40.76%), ECCI ($CO_{2}/E$, 11.32%), and population (P, 10.78%) are the increasing factors of carbon emissions; STIIE (GDP/T, −87.57%), OCI ($CO_{2}/GDP$, −35.84%), EI (E/GDP, −9.77%), and STIICI ($CO_{2}/T$, −1.64%) are the decreasing factors.

Secondly, under the baseline development scenario, the carbon peak goal cannot be achieved for Xi’an. The economic growth of Xi’an led to the rapid growth of carbon emissions during 1995–2020. It grew from 899.12 × 10^4^ tons in 1995 to 4912.14 × 10^4^ tons in 2020. The simulation results show that the uncertainty of carbon emission increases under this scenario. By 2060, the 95% confidence interval for carbon emissions will increase to 30,090.25–155,903.42 × 10^4^ tons, and Xi’an will fail to achieve the carbon peak goal by 2030.

Thirdly, under the green development scenario, the carbon emissions of Xi’an will peak by 2051. Under this scenario, the carbon emissions of Xi’an will peak in 2051, with a 95% confidence interval of 6668.47–7756.90 × 10^4^ tons. The rapid growth of carbon emissions could be effectively restrained, while the government takes active energy conservation and emission reduction measures. The STII to improve production efficiency and expand production scale accounts for a relatively large proportion, and it makes the carbon emissions fail to peak by 2030.

Finally, under the technology breakthrough scenario, the carbon peak goal could be achieved by 2030. Under this scenario, the lower and median boundaries of carbon emissions peak at 4703.94 × 10^4^ tons and 4852.39 × 10^4^ tons in 2026; the upper boundary peaks at 5042.15 × 10^4^ tons in 2030. According to the EKC theory, carbon emissions will peak between 2028 and 2029 in Xi’an. However, it would fail to achieve the carbon neutrality goal by 2060 and should rely on the national carbon trading market of China to achieve the goal with a trading volume of 2524.61–3007.01 × 10^4^ tons.

## Figures and Tables

**Figure 1 ijerph-19-15039-f001:**
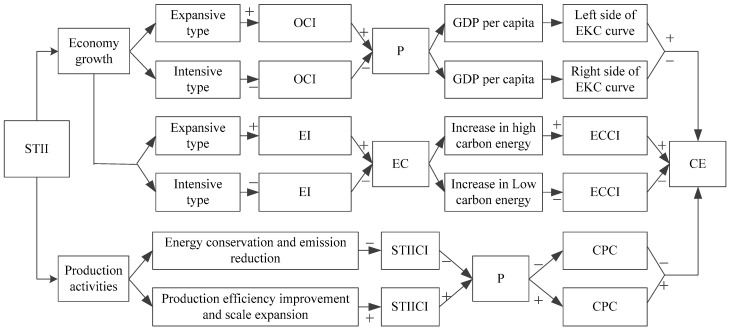
Mechanism of STI influencing carbon emissions.

**Figure 2 ijerph-19-15039-f002:**
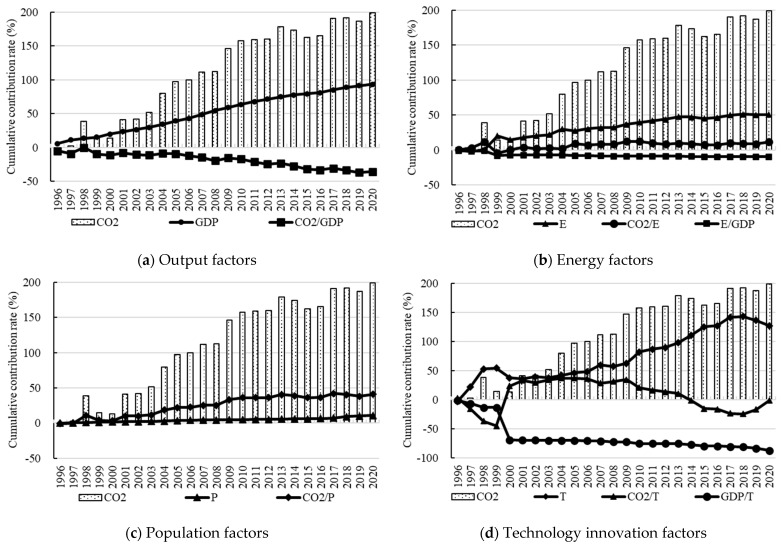
Cumulative effect of different factors.

**Figure 3 ijerph-19-15039-f003:**
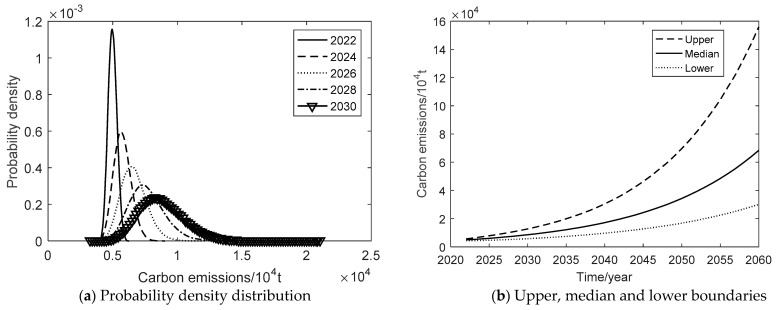
Carbon emission of the baseline development scenario.

**Figure 4 ijerph-19-15039-f004:**
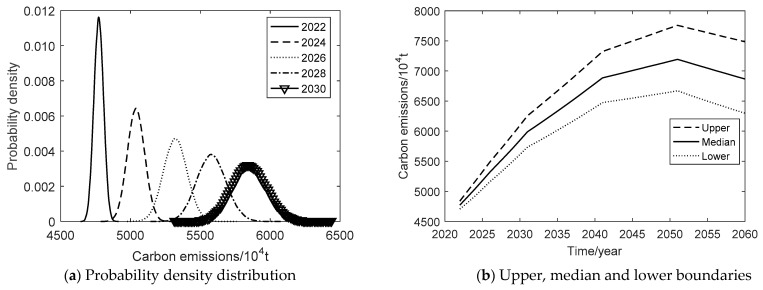
Carbon emission of the green development scenario.

**Figure 5 ijerph-19-15039-f005:**
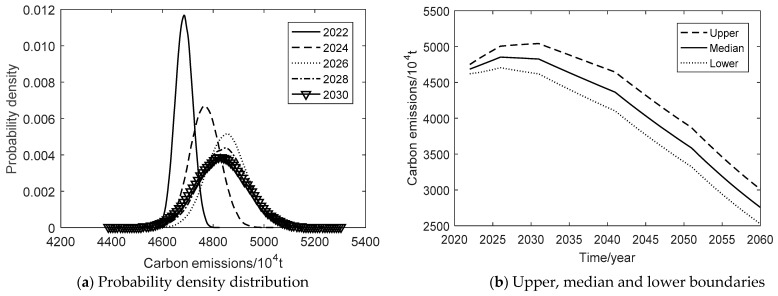
Carbon emission of the technology breakthrough scenario.

**Figure 6 ijerph-19-15039-f006:**
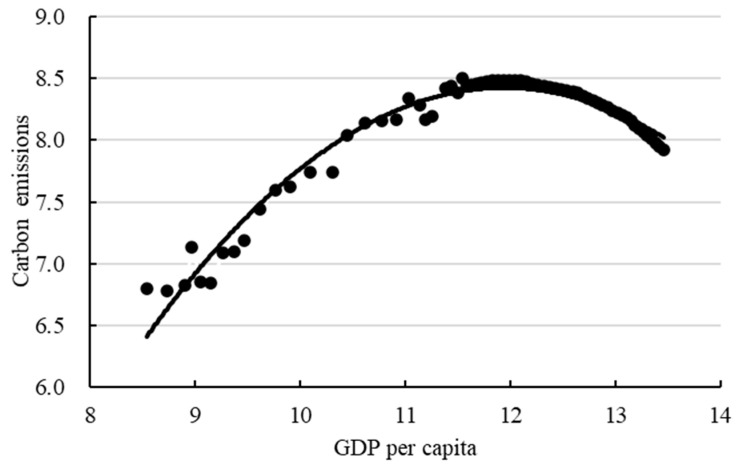
Relationship between carbon emissions and GDP per capita in Xi’an.

**Table 1 ijerph-19-15039-t001:** Literature on decomposition methods and driving factors of carbon emissions.

Literature	Research Fields	Period	Method	Factors
Pan et al. (2018) [21]	The northeast, central region, west, and coastal region of China	2002–2010	SDA	Carbon intensity, production technology, final demands (investment and consumption), exports
Wang et al. (2019) [5]	The Beijing-Tianjin-Hebei region of China	2002–2012	SDA	Carbon intensity, intermediate demand, consumption structure, consumption level, population
Zhengnan et al. (2014) [35]	Eight major industry sectors of China	2003–2011	LI	Structural factors, efficiency factors
Chen et al. (2022) [22]	Coal consumption in China	2005–2017	LI	Economic growth, coal intensity
Mohammad and Wu (2020) [23]	Electricity sector of Bangladesh	1979–2018	LMDI	Carbon intensity, substitutions, energy intensity, GDP per capita, population
Alajmi (2021) [24]	Greenhouse gas in Saudi Arabia	1990–2016	LMDI	GDP, energy consumption, population
Roux and Plank (2022) [25]	Energy use in the USA	1995–2016	LMDI	Economic output (GDP), energy intensity, share of sector
Vaninsky (2014) [28]	The United States and China	1980–2012	GDIM	GDP, energy consumption, population, intensity of energy, energy intensity of economic activity, GDP per capita, CO_2_ per capita
Shao et al. (2017) [36]	The manufacturing sector of China	1995–2014	GDIM	GDP, GDP carbon intensity, energy use, energy structure, energy intensity, investment, investment carbon intensity, investment intensity
Li et al. (2020) [15]	The construction industry in China	2001–2017	GDIM	GDP, carbon intensity of output, energy consumption, energy consumption intensity, labor population, carbon intensity of energy consumption, labor productivity per capita, carbon emissions of the labor force, construction industry labor share, construction industry labor productivity

**Table 2 ijerph-19-15039-t002:** The acronyms and variables involved in GDIM.

Acronyms	Meaning	Variables
CE	Carbon emissions	$Z=CO_{2}$
GDP	Economic output (GDP)	$X_{1}=GDP$
OCI	Output carbon intensity	$X_{2}=CO_{2}/GDP$
EC	Energy consumption	$X_{3}=E$
ECCI	Energy consumption carbon intensity	$X_{4}=CO_{2}/E$
STII	Science and technology innovation investment	$X_{5}=T$
STIICI	Science and technology innovation investment carbon intensity	$X_{6}=CO_{2}/T$
P	Population	$X_{7}=P$
CPC	CO_2_ per capita	$X_{8}=CO_{2}/P$
STIIE	Science and technology innovation investment efficiency	$X_{9}=GDP/T$
EI	Energy intensity	$X_{10}=E/GDP$

**Table 3 ijerph-19-15039-t003:** The annual rate of each factor in the baseline development scenario (%).

Factors	2021	2022–2060		
		Min	Med	Max
*T*	24.76	−1.77	11.27	25.54
*GDP/T*	−14.50	−8.69	1.94	13.05
*E/GDP*	−5.11	−7.64	−6.79	−6.03
*CO_2_/E*	−6.60	−0.64	1.11	2.74

**Table 4 ijerph-19-15039-t004:** The annual rate of each factor in the green development scenario (%).

Factors	2021	2022–2025			2026–2030			2031–2040			2041–2050			2051–2060		
		Min	Med	Max	Min	Med	Max	Min	Med	Max	Min	Med	Max	Min	Med	Max
*T*	24.76	7.00	8.00	9.00	6.00	7.00	8.00	5.00	6.00	7.00	4.00	5.00	6.00	3.00	4.00	5.00
*GDP/T*	−14.50	−2.00	−1.00	0.00	−1.47	−0.47	0.53	−1.47	−0.47	0.53	−1.48	−0.48	0.52	−1.48	−0.48	0.52
*E/GDP*	−5.11	−3.86	−2.86	−1.86	−3.86	−2.86	−1.86	−3.86	−2.86	−1.86	−3.86	−2.86	−1.86	−3.86	−2.86	−1.86
*CO* _2_ */E*	−6.60	−1.26	−1.06	−0.86	−1.22	−1.02	−0.82	−1.25	−1.05	−0.85	−1.25	−1.05	−0.85	−1.25	−1.05	−0.85

**Table 5 ijerph-19-15039-t005:** The annual rate of each factor in the technology breakthrough scenario (%).

Factors	2021	2022–2025			2026–2030			2031–2040			2041–2050			2051–2060		
		Min	Med	Max	Min	Med	Max	Min	Med	Max	Min	Med	Max	Min	Med	Max
*T*	24.76	7.00	8.00	9.00	6.00	7.00	8.00	5.00	6.00	7.00	4.00	5.00	6.00	3.00	4.00	5.00
*GDP/T*	−14.50	−3.00	−2.00	−1.00	−2.47	−1.47	−0.47	−2.47	−1.47	−0.47	−2.48	−1.48	−0.48	−2.48	−1.48	−0.48
*E/GDP*	−5.11	−4.66	−3.66	−2.66	−4.66	−3.66	−2.66	−4.66	−3.66	−2.66	−4.66	−3.66	−2.66	−4.66	−3.66	−2.66
*CO* _2_ */E*	−6.60	−1.26	−1.06	−0.86	−1.85	−1.65	−1.45	−1.81	−1.61	−1.41	−1.81	−1.61	−1.41	−1.81	−1.61	−1.41

**Table 6 ijerph-19-15039-t006:** The average annual rates in different stages of three scenarios.

Scenario	Boundary	2022–2025	2026–2030	2031–2040	2041–2050	2051–2060
Baseline development scenario	Upper	11.98	9.88	9.04	8.59	8.39
	Median	7.13	7.13	7.14	7.13	7.12
	Lower	2.56	4.48	5.31	5.68	5.96
Green development scenario	Upper	3.23	2.67	1.59	0.58	−0.39
	Median	2.76	2.39	1.40	0.44	−0.52
	Lower	2.29	2.12	1.22	0.30	−0.64
Technology breakthrough scenario	Upper	1.35	0.16	−0.81	−1.80	−2.76
	Median	0.88	−0.11	−1.00	−1.95	−2.88
	Lower	0.41	−0.38	−1.19	−2.08	−3.00

## Data Availability

Data of economic and environment indicators involved in this paper can be obtained from the Xi’an Social Statistics Yearbook, the Shaanxi Social Statistics Yearbook, the Shaanxi Science and Technology Statistics Yearbook and the China Energy Statistics Yearbook.

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
