# Peer review of "Dynamic Scenario Analysis of Science and Technology Innovation to Support Chinese Cities in Achieving the “Double Carbon” Goal: A Case Study of Xi’an City"

_ijerph, 2022, doi:10.3390/ijerph192215039_

Round 1
Reviewer 1 Report
This paper is interesting and the data is very useful. But is the tranlation of "double carbon" suitable?
Author Response
Thank you for your comments concerning our manuscript entitled “Dynamic scenario analysis of science and technology innovation to support Chinese cities in achieving the “double carbon” goal: A case study of Xi’an city” (Manuscript ID: ijerph-1926335). Please refer to the attachment for the detailed revision of the paper.

Reviewer 2 Report
Focusing on the impact of STI on the carbon emission, negative or positive, this article clarifies the main influencing factors and their contributions through GDIM model. The conclusions are credible and reasonable, and could be the indicator of implementation for achieving the “double carbon” goals in Xi’an city.
Author Response

(The authors gave the same response as above.)

Reviewer 3 Report
The manuscript entitled: "Dynamic scenario analysis of science and technology innovation to support Chinese cities in achieving the “double carbon” goal: A case study of Xi'an city" is clearly presented, however, some considerations need to be made.
- Line 32: remove the president's name;
- Lines 121 and 141: put "Table" in capital letters;
- Line 307: highlight item 3.3.3;
- Figure 5 comes before its call in the text;
- Line 342: separate "Figure2";
- Figures 6 and 7 must come before Table 6;
- Increase the size of the Figures to allow better visualization and put your call in bold text.
The manuscript has 12 figures and 7 tables. A very high number. Place figures in A, B, C... or add material to the attachment.
Author Response

(The authors gave the same response as above.)
